# A teacher-teacher framework for clinical language representation learning

**Feiqing Huang**[*]
Harvard T.H. Chan School of Public Health
fqhuang@hsph.harvard.edu

**Shenghan Zhang**[*]
Harvard Medical School
shenghan_zhang@hms.harvard.edu

**Sara Morini Sweet**
Harvard Medical School
sara_morini@hms.harvard.edu

**Tianxi Cai**[†]
Harvard T.H. Chan School of Public Health
Harvard Medical School
tcai@hsph.harvard.edu

## Abstract

In recent years, there has been a proliferation of ready-to-use large language models (LLMs) designed for various applications, both general-purpose and domain-specific. Instead of advocating for the development of a new model or continuous pretraining of an existing one, this paper introduces a pragmatic teacher-teacher framework to facilitate mutual learning between two pre-existing models. By leveraging two teacher models possessing complementary knowledge, we introduce a **LI**ghtweight k**N**owledge alignm**E**nt (LINE) module aimed at harmonizing their knowledge within a unified representation space. This framework is particularly valuable in clinical settings, where stringent regulations and privacy considerations dictate the handling of detailed clinical notes. Our trained LINE module excels in capturing critical information from clinical notes, leveraging highly de-identified data. Validation and downstream tasks further demonstrate the effectiveness of the proposed framework.

## 1 Introduction

Clinical data frequently appears in various forms, with each format capturing overlapping but complementary aspects of patient information. For instance, narrative details in a clinical note may be processed into a structured list of clinical concepts for research purposes. Similarly, a CT scan report provides descriptive findings that are directly linked to the visual data in the corresponding scan. Ideally, both forms of data would be jointly accessible; however, practical challenges, such as privacy restrictions, often limit us to only one form. This limitation raises an important question:

*Could we use the accessible form of data to approximate or represent the unobserved one?*

To explore this question, we can leverage the capabilities of large pretrained models. Recent years have seen the emergence of large language models (LLMs), which have demonstrated impressive performance across a range of tasks, including prediction, generation, and representation learning [1, 27, 34]. In the clinical domain, many pretrained LLMs—such as CODER [38], UmlsBERT [15], Clinical BioBERT [2], SapBERT [13], PubmedBERT [8], and BioBERT [11]—are based on BERT architectures with approximately 300 million parameters. These models are typically initialized with BERT weights and then undergo further pretraining on biomedical texts, a process often referred to

---

[*]Equal contribution.
[†]Correspondence: tcai@hsph.harvard.edu

38th Conference on Neural Information Processing Systems (NeurIPS 2024).

as continual pretraining. While this approach allows the model to integrate both general and domain-specific knowledge, it is also time-consuming and resource-intensive. The recent proliferation of off-the-shelf clinical LLMs now prompts a pragmatic idea:

*Can the pretrained models be made to directly exchange knowledge with one another?*

Inspired by the two questions posed above, we propose a teacher-teacher framework for clinical representation learning. The core idea is as follows: given paired training data in two distinct forms and two pretrained models with complementary knowledge bases, the framework processes each form of data separately through one of the models. It then processes the resulting representations through a newly proposed module and further aligns them within a unified representation space. This alignment entails a mutual teaching process, where both models act as teachers, exchanging their knowledge. The complementary knowledge arises from two sources: first, the complementary information inherent to each form of data, and second, the distinct knowledge embedded in each pretrained model due to differences in their original training corpora. We summarize our key innovations as follows:

- **LINE module for teacher-teacher learning**: We propose a lightweight knowledge alignment (LINE) module to facilitate effective teacher-teacher learning. This module takes representations from two pretrained teacher models and projects them into a unified representation space, guided by an alignment loss and a relational loss.

- **Efficient training with residual information recovery**: We adopt a two-stage, few-epoch training process focused solely on the LINE module, keeping the pretrained model weights frozen. In the first stage, we train the LINE module for a few epochs to achieve initial alignment and to capture residuals (complementary information across two forms of data). In the second stage, these residuals are used to refine the training data for further few-epoch training, offering insights into the complementary information between data forms.

- **Generating cross-form representation**: Once trained, the LINE module enables cross-form representation for downstream tasks. When only one data form is available, the corresponding teacher model and LINE projection can generate a proxy for the missing data, enabling its use in the downstream tasks.

We evaluate our teacher-teacher framework on two fronts: (1) whether knowledge alignment can improve the performance of one or both teacher models across different tasks, and (2) whether the available data form can be used to generate better proxies for the other, missing data form.

## 1.1 Clinical use case

While using one form data to represent another is a relevant question across various fields, we demonstrate the utility of our proposed method through a real-world clinical use case. This example is particularly relevant to clinical data privacy concerns, highlighting its practical importance. Specifically, we focus on two data forms: clinical notes and their corresponding structured lists of clinical concepts. Clinical notes play a crucial role in numerous applications, such as disease diagnosis, disease progression prediction, patient classification, and risk assessment [25, 29, 14].However, working with real clinical notes is subject to strict regulations due to privacy concerns. Sharing clinical notes between institutions necessitates approval from an institutional review board (IRB), adding layers of regulatory complexity and substantially extending project timelines.

These challenges are a major barrier to scaling up clinical datasets for pre-training. As an alternative, efficient software [e.g. 36, 23] can directly extract structured clinical concepts from clinical notes. This approach offers two advantages: First, because clinical concepts are drawn from a predefined dictionary – specifically, the Unified Medical Language System (UMLS) [5] – it reduces the risk of including sensitive information, making data sharing across research groups more feasible. Second, known relationships among many clinical concepts can be further incorporated to form a structured graph, capturing both the textual content and interrelationships among concepts.

In this use case, we treat the clinical note as the missing data form and reframe our question accordingly: Can we leverage the list of clinical concepts to effectively summarize and provide a proxy for the content in the clinical note?

## 1.2 Related works

**Different modes of learning** Existing knowledge within pretrained LLMs varies based on their learning modes and domains of their pretrained data. The first mode, learning by generalization, involves training models on extensive datasets to generalize knowledge through example-based learning [1, 27]. These models encode knowledge directly into their embeddings, which capture syntactic and semantic relationships in texts. Numerous studies have demonstrated the efficacy of LLMs in encoding syntactic heuristics and retaining factual knowledge to varying extents [7, 19, 28, 18, 21]. Recent advancements, exemplified by models such as Sentence-T5 [17], E5 [31], BGE [34], and GPT-4 [1], have focused on generating high-quality textual representation for downstream tasks. However, in specialized clinical domains, vocabulary diverges significantly from general corpora, requiring that these general-purpose models undergo additional training on biomedical data to ensure relevance and accuracy. Models like BioBERT [11] and PubmedBERT [8], pretrained on biomedical texts, have demonstrated superior performance compared to BERT in biomedical-specific tasks. Clinical BioBERT [2] further augment their training data with MIMIC datasets to incorporate medical knowledge derived from real-world medical practices. Nevertheless, scaling up training data in the clinical domain poses challenges due to privacy regulations, as previously noted.

The second mode, learning by integrating pre-existing knowledge, focuses on encoding established factual associations between concepts into the model's representation space. In the clinical domain, knowledge repositories such as UMLS [5] offer comprehensive biomedical ontologies and relatedness information between concepts, which can be treated as existing knowledge graph and harnessed during the pre-training phase. For example, SapBERT [13] leverages synonymous relationships extracted from UMLS during self-supervised pretraining, strengthening the model's understanding of semantic similarity. UmlsBERT [15] incorporates semantic type information from UMLS, enabling a deeper semantic understanding of clinical texts. Furthermore, CODER [38] encodes more granular relationships, such as "may prevent or treat" and "may cause", into their knowledge networks, enhancing its ability to capture medical associations. By integrating such pre-existing domain-specific knowledge graph during training, these models show improved ability to navigate the intricacies of specialized domains, leading to better performance in domain-specific tasks [13, 15, 38].

Distinct from these two modes, the proposed teacher-teacher framework adopts the third mode, learning by alignment. This mode of learning involves creating a joint representation space between two models using paired data, such as an image with its textual description, audio/video with its transcript, or multilingual translated texts. This approach focuses on teaching models to distinguish between paired and unpaired data, making it especially effective for multimodal applications. For example, the seminal CLIP framework [20] uses different encoders for text and images to embed each mode of information separately. It then employs a contrastive loss to align these multimodal embeddings by end-to-end training on both encoders, ensuring that related text and image pairs are closely aligned in the representation space. Similarly, Li et al. [12] utilize a text encoder and a text-referred audio encoder with inter-modality and intra-modality training, enhancing the model's ability to integrate complementary features from different views. Wang et al. [33] propose an efficient global-local alignment framework for co-training language with videos, and its improved performance further highlights the advantages of learning the homogeneity as well as heterogeneity across language and visual modalities. For multilingual translation tasks, alignment learning has also demonstrated remarkable efficacy. For example, Reimers and Gurevych [22] introduce a teacher-student network model in which a fixed teacher network, well-versed in a target language, guides a student network, initially proficient in a different language, to emulate the teacher's knowledge. This method not only enhances translation accuracy but also creates a joint representation space that offers potentially interesting insights into cross-linguistic differences.

**Other related works** In a broader context, our approach is remotely related to generative domain adaptation; however, unlike generative domain adaptation, our method requires paired data during training. Traditional methods in this field often utilize generative models to synthesize samples across domains, helping the model generalize effectively in new contexts. Some example frameworks include generative adversarial networks [40], variational autoencoders [32], diffusion models [4], or generative pretrained transformers [35]. However, our framework diverges from these in that we specifically deploy two pretrained teacher models, each with distinct expertise and capabilities. This design allows the models not only to align but also to exchange complementary knowledge during the learning process, potentially enhancing the performance of one or both teacher models.

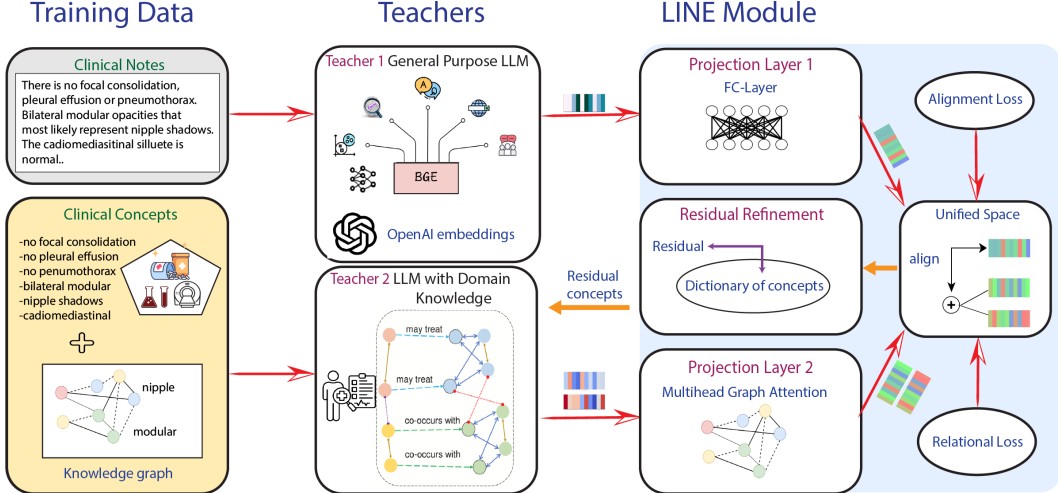

Figure 1: The proposed teacher-teacher framework is illustrated for a specific use case. The two forms of training data are unstructured clinical notes and structured clinical concepts represented in a relational graph. Two pretrained teacher LLMs independently embed these two forms of data, with the LINE module facilitating mutual learning and alignment. Through this process, the embeddings of the two data forms converge toward a shared representation space, enabling knowledge exchange and cross-form representation between the two teacher models.

## 2 The teacher-teacher framework

### 2.1 Training data and alignment objective

This section provides a detailed illustration of our LINE framework for a specific clinical use case, aligning clinical concepts with corresponding clinical texts. The algorithm can be extended to support a broad range of other alignment tasks, as discussed in Section 4. Specifically, our paired training data take the form (CLINICAL TEXT, LIST OF CLINICAL CONCEPTS), where each list of clinical concepts is derived from the clinical texts. We further augment the clinical concepts with known relationships from UMLS to form a structured graph. The primary objective is to create a "summarized" embedding for the list of concepts and align it with the projected text embedding, as illustrated in Figure 1.

### 2.2 Teacher models

A key design principle in our framework is selecting two teacher models with complementary knowledge bases. To this end, we choose (1) a general-purpose LLM pretrained on large-scale, general-domain text, and (2) a domain-specific LLM with established knowledge of biomedical concept relationships. In this setup, the general-purpose model (Teacher 1) learns factual relationships between clinical concepts from the domain-specific model (Teacher 2), while Teacher 2 enhances its grasp of the contextual connections linking these concepts. In the paper, we explore two combinations of teacher models: BGE+CODER and GPT-4+CODER.

**Teacher 1: General purpose LLM** BGE is a general-purpose embedding model trained on large, diverse language datasets. Initially trained on 100 million text pairs from open web corpora, BGE further underwent multitask learning on various retrieval and reranking tasks, including the CMedQA-v2 medical question and answer dataset [39]. It ranks highly on the Massive Text Embedding Benchmark (MTEB) dataset [16], which covers tasks like classification, reranking, retrieval, STS, and paired classification tasks. GPT-4 is a large-scale, multimodal model considered as one of the state-of-the-art tools for language modeling and generation. Although its model architecture and trained weights are not publicly available, GPT-4 can generate embeddings for textual data, accessible through the OpenAI API. For our experiments, we used the "text-embedding-3-small" model, which produces embeddings with a dimension of 1536.

**Teacher 2: LLM with domain knowledge** CODER is initialized by PubMedBERT [8] and trained on the UMLS 2020AA release relation triplets, encompassing 4.27M concepts, 15.48M terms, and 87.89M relations across 127 semantic types, 14 relation types, and 923 relationship attributes. It employs contrastive learning with relation triplets, i.e. (*head entity*, *relation*, *tail entity*). CODER demonstrates superior performance in term normalization and disease classification tasks compared to SapBERT, Clinical BERT, PubMedBERT, and BioBERT.

In our two combinations of teacher models, the general purpose LLMs (BGE and GPT-4) with their extensive and diverse training, generally outperform CODER. This imbalance might suggest that they cannot learn from CODER. However, our experiments suggest that not only can CODER benefit from the knowledge of the general-purpose LLM, but the performance of BGE or GPT-4 also improves on certain tasks with alignment training. We hypothesize that the alignment process regularizes the embedding space of the general-purpose LLM by incorporating CODER's association graph, resulting in a more diversified and robust representation space.

## 2.3 LIghtweight kNowledge alignmEnt module (LINE)

Our framework eliminates the need for any further training of the original teacher models. Instead, we introduce a trainable LIghtweight kNowledge alignmEnt module (LINE), which explicitly models projection operations to align the embeddings from the general-purpose LLM and the domain-specific LLM into a unified representation space.

First, we derive clinical text embeddings and concept embeddings from the general-purpose LLM and domain-specific LLM using pretrained checkpoints available through Huggingface or the OpenAI API, denoted as $\mathbf{t}^{(0)} \in \mathbb{R}^{d_t}$ and $\mathbf{c}^{(0)} \in \mathbb{R}^{d_c}$, respectively. These initial embeddings retain the original knowledge encoded within each teacher model. We then incorporate trainable projection layers specific to each LLM to refine and align these embeddings.

**Projection layer for general purpose LLM: Fully-connected layer** The projection layer for the output embeddings $\mathbf{t}^{(0)}$ from BGE or GPT-4 is implemented as a simple fully-connected linear layer:

$$\mathbf{t} := \mathcal{F}(\mathbf{t}^{(0)}) = \text{FC}(\mathbf{t}^{(0)}).$$

This approach aligns with prior work [3, 6, 20], where linear projections are used without non-linear activation functions.

**Projection layer for domain-specific LLM: Multihead graph attention** This layer is designed to reinforce the domain-specific knowledge embedded in CODER from the UMLS ontology. We focus on modeling seven broad categories of UMLS relations: *parent-child hierarchy*, *synonyms*, *related and possibly synonymous*, *broad relationship*, *narrow relationship*, *non-synonymous/narrow/broad relationship*, and *quantifiable relationships*. Each relationship type is represented by a distinct adjacency matrix, and a multi-head graph attention (MHGA) layer assigns each type to a separate attention head. This design enables the layer to explicitly account for the unique structure of each relation. The outputs from all attention heads are aggregated and projected into the joint representation space. Formally, for any concept with an initial CODER embedding $\mathbf{c}^{(0)}$, we have:

$$\mathbf{c} := \mathcal{G}(\mathbf{c}^{(0)}) = \text{average}_{k \in [7]} \left[ \text{GraphAttn}_k(\mathbf{W}_c \mathbf{c}^{(0)}, \text{adjacency matrix}_k) \right], \tag{1}$$

where $\mathbf{W}_c \in \mathbb{R}^{d_t \times d_c}$ is the projection matrix that maps the initial CODER embeddings to the same dimension as the BGE or GPT-4 embeddings. Additional details on the handling of negatively mentioned concepts are provided in the Appendix.

## 2.4 Training strategy

To enable the two teacher models to enhance each other's knowledge in a time-efficient and resource-efficient way, we adopt a two-stage, few-epoch training strategy exclusively for the LINE module. This approach effectively aligns the embeddings from the general-purpose and domain-specific LLMs without any additional training of the original pretrained models. It also allows for the use of proprietary models, such as GPT-4, where access to the underlying pretrained model is restricted.

**Unified representation via alignment** The first stage involves an initial round of few-epoch training aimed at defining residuals, i.e. the differences between the learned representations of the two data forms. This stage is critical for capturing complementary information between the two models.

Specifically, we aim to align the embedding for clinical texts with its list of associated concept embeddings in the joint representation space. This alignment is quantified using a monotonic score function $s$, which measures the degree of alignment between the elements of each data pair. Formally, let $\mathbf{p}_i = (\{\mathbf{c}_{i,j}\}_{j\in[m_i]}, \mathbf{t}_i)$ denote a positive pair, where $\mathbf{t}_i$ is the clinical text embedding and $\{\mathbf{c}_{i,j}\}_{j\in[m_i]}$ are the corresponding concept embeddings with a total of $m_i$ concepts, for all $i \in [N]$. The negative pair is defined as $\mathbf{n}_{i,i'} = (\{\mathbf{c}_{i,j}\}_{j\in[m_i]}, \mathbf{t}_{i'})$ for $i' \neq i \in [N]$. The score function $s$ is defined as $s(\cdot) = s(\{\mathbf{c}_j\}_{j\in[m]}, \mathbf{t}) = -\|\mathbf{t} - \text{mean}(\{\mathbf{c}_j\}_{j\in[m]})\|_2$. Our goal is to ensure $s(\mathbf{p}_i) \geq s(\mathbf{n}_{i,i'})$, indicating better alignment of positive pairs over negative pairs.

To account for complementary information in the positive pairs, we introduce a data-dependent alignment weight $\rho_i \in (0, 1]$ to allow varying degrees of information overlap. For example, in extracting clinical concepts from clinical texts, misspellings can result in failure to detect key concepts; see Table A1 in the Appendix for concrete examples. A smaller $\rho_i$ value indicates more tolerance for misalignment, reflecting cases with less overlap between the text and concept embeddings. The alignment loss is defined as:

$$\sum_i \sum_{i' \neq i} \rho_i f(s(\mathbf{p}_i), s(\mathbf{n}_{i,i'})),$$

where $f$ is a contrastive loss function, and we used the triplet loss [24] for our experiments. In our specific context, the weight $\rho_i$ is computed as the percentage of words in the text captured in the concept list, i.e. $\rho_i = \frac{\text{number of words accounted for in text } i}{\text{total word count in text } i}$.

In addition, to preserve the integrity of existing concept relations, we incorporate a relational contrastive loss during training. Here, positive pairs represent known relationships, while negative pairs indicate absent or unknown relationships. This additional loss term ensures that information from the known association graph is retained throughout the alignment process.

**Refinement through residual recovery** Once the first training stage stabilizes, we move to the second stage, focusing on recovering residual information to further refine the alignment. The residual $\mathbf{e}_i$ for each data pair is computed as:

$$\mathbf{e}_i = \mathbf{t}_i - \text{mean}(\{\mathbf{c}_{i,j}\}_{j\in[m_i]}), \quad i \in [N].$$

To further consolidate information from residuals, we make use of a projected concept dictionary $\mathbb{C}_{\mathcal{G}}$, consisting of all clinical concepts from UMLS, projected using the trained $\mathcal{G}$ defined in (1). For each residual $\mathbf{e}_i$, we select concepts from $\mathbb{C}_{\mathcal{G}}$ that have a cosine similarity of 0.9 or higher with $\mathbf{e}_i$. The most similar concept from the selected subset, denoted as $\mathbf{c}_{i,m_i+1}$, is then added to the concept embeddings set, refining the training pair to $(\{\mathbf{c}_{i,j}\}_{j\in[m_i+1]}, \mathbf{t}_i)$ for further few-epoch training.

## 3  Training and validation

### 3.1  Data preprocessing and training

Our model, referred to as LINE, is trained with two configurations of teacher models: BGE+CODER and GPT-4+CODER. For our training dataset, we utilized 332K discharge notes and 2.2M radiology reports from 146K patients available in the MIMIC-IV database [10]. Given the structured nature of both types of clinical notes, we began by segmenting the notes into their respective sections. Radiology reports were divided into seven sections: examination, indication, technique, comparison, finding, procedures, and impression. Similarly, discharge notes were segmented into fourteen sections: chief complaint, major surgical or invasive procedure, history of present illness, past medical history, social history, family history, physical exam, pertinent results, brief hospital course, medications on admission, discharge medications, discharge disposition, discharge diagnosis, discharge condition, discharge instructions, and follow-up instructions.

To extract clinical concepts from these notes, we employed the Narrative Information Linear Extraction (NILE) software [36]. NILE outputs a list of clinical concepts present in each segmented text, indicating whether each concept is mentioned positively or negatively. Figure 1 provides an illustrative example. Next, we combined the original text with its extracted clinical concepts to create paired training data. This preprocessing resulted in a dataset of 7.2M text-concept pairs. During this process, we located the extracted concepts within the text and counted the number of words accounted for by the list of concepts. The alignment weight for each text was then calculated as $\rho = \frac{\text{number of words accounted for}}{\text{total word count}}$. To construct the relational adjacency matrices for the CODER projection

Table 1: For each positive concept-text pair, we randomly selected 100 other texts to replace the original text, forming negative pairs. We then calculated the cosine similarity between the mean of the clinical concept list and the text, with embeddings generated by various models. By ranking these cosine similarities from highest to lowest, we identified the rank of each positive pair among its negative pairs and computed the mean rank, mean reverse rank, and Top-10 accuracy (Top10@Acc). Improvements over the corresponding teacher models are indicated by "↓" for mean rank and "↑" for mean reverse rank and Top10@Acc.

| Models / Metric | Mean Rank ↓ | Mean Reverse Rank ↑ | Top10@Acc ↑ |
|---|---|---|---|
| BioBERT | 35.83 | 0.127 | 0.258 |
| Clinical BioBERT | 34.51 | 0.143 | 0.287 |
| SapBERT | 15.49 | 0.346 | 0.588 |
| PubMedBERT | 39.55 | 0.117 | 0.210 |
| CODER | 43.79 | 0.092 | 0.181 |
| BGE | 5.526 | 0.534 | 0.845 |
| GPT-4 | 1.778 | 0.820 | 0.988 |
| LINE | | | |
|   BGE+CODER | 1.437↓ | 0.873↑ | 0.997↑ |
|   GPT-4+CODER | 1.477↓ | 0.872↑ | 0.995↑ |

layer, we leveraged the 2022AB release of the UMLS knowledge graph, which encompasses 9M concepts, defined by 4M unique identifiers (CUIs), and includes 25M relationships.

We adopted the Adam optimizer with a learning rate of $10^{-3}$ and a batch size of 128. The training process was divided into two stages as detailed in Section 2.4: We first trained the model for three epochs initially, during which we identified residual concepts not adequately captured. Then, using the model checkpoint from the first phase, we recovered the residual concepts and refined the training pairs. The model was then trained for an additional two epochs. The alignment loss and the relational loss both converged rapidly. All experiments were conducted using an NVIDIA RTX 8000 GPU with 48GB of VRAM. The entire training process required less than 10 hours on a single GPU.

## 3.2 Validation on alignment objective

This section validates the effectiveness of our trained LINE model in aligning the representation spaces of the two teachers. To achieve this, we utilized a holdout subset of patients whose clinical notes were excluded from the training set. This subset comprised 100K radiology reports from 92K patients. We applied the same data preprocessing steps, resulting in 275K text-concept pairs. Our evaluation method involved the following steps. For each positive pair, we randomly sampled 100 negative pairs by substituting the clinical text with a different one. We then computed the cosine similarity between the mean embedding of the concept list and the text embedding for all pairs, in line with our alignment objective. Finally, we ranked these pairs based on their cosine similarity scores, from highest to lowest.

To assess performance, we calculated the mean rank, mean reverse rank, and Top-10 accuracy for all positive pairs. These metrics are presented in Table 1. We compared our LINE model against several models: GPT-4, BGE, CODER, SapBERT, Clinical BioBERT, PubMedBERT, and BioBERT. As shown in Table 1, LINE with BGE+CODER achieves the highest alignment between concept and text representations, outperforming the other models. Both LINE models show improved alignment compared to their respective teacher models, validating the consistency of our training objectives.

## 3.3 Validation on downstream tasks

In this section, the performance of our trained LINE models on various downstream tasks is evaluated based on two main aspects: (1) to determine whether the teacher models can benefit from mutual learning, and (2) to assess whether the LINE module can generate more effective proxies for clinical text using the list of clinical concepts. Specifically, we use the LINE-projected CODER embeddings for tasks involving clinical concepts and the LINE-projected BGE or GPT-4 embeddings for tasks involving clinical text. Given that two model configurations are used, we refer to each configuration as "LINE" in our tables, with "Teacher 1+Teacher 2" specified for clarity. Three types of downstream

Table 2: AUCs for detecting related pairs versus randomly selected pairs under various models. The classes of clinical concepts include parent-child hierarchy, siblings hierarchy, may treat or may prevent, classifies as, differential diagnosis, method of, and causative of. The last row lists the detailed number of positive pairs in each class. Improvements over the corresponding teacher models are indicated by "↑".

| Models / Relation | Parent | Sibling | May Treat/Prevent | Classifies | DDX | Method of | Causative |
|---|---|---|---|---|---|---|---|
| BioBERT | 0.785 | 0.754 | 0.442 | 0.842 | 0.665 | 0.722 | 0.708 |
| Clinical BioBERT | 0.837 | 0.814 | 0.386 | 0.885 | 0.733 | 0.825 | 0.753 |
| SapBERT | 0.940 | 0.874 | 0.672 | 0.960 | 0.874 | 0.871 | 0.914 |
| PudmedBERT | 0.637 | 0.651 | 0.639 | 0.640 | 0.634 | 0.535 | 0.604 |
| CODER | 0.943 | 0.886 | 0.441 | 0.969 | 0.844 | 0.774 | 0.899 |
| BGE | 0.968 | 0.940 | 0.840 | 0.984 | 0.927 | 0.906 | 0.944 |
| GPT-4 | 0.974 | 0.940 | 0.825 | 0.991 | 0.939 | 0.934 | 0.935 |
| LINE | | | | | | | |
|   BGE+CODER | 0.978 ↑ | 0.939 | 0.926 ↑ | 0.989 ↑ | 0.959 ↑ | 0.971 ↑ | 0.953 ↑ |
|   GPT-4+CODER | 0.977 ↑ | 0.932 | 0.931 ↑ | 0.988 | 0.938 | 0.965 ↑ | 0.947 ↑ |
| # relation pairs | 125152 | 48252 | 14686 | 14073 | 9777 | 6588 | 3711 |

Table 3: The mean of the F1 scores for different models over five random initializations on two i2b2 datasets. The results for BioBERT, Clinical BioBERT and UmlsBERT were directly copied from [15] for comparison. Here, the LINE projection is applied to the token-level BGE embeddings. The best result under each metric is highlighted in bold font.

| Dataset | i2b2 2006 | | i2b2 2014 | |
|---|---|---|---|---|
| | Test. F1 | Val. F1 | Test. F1 | Val. F1 |
| BioBERT | 93.3 | 93.8 | 94.6 | 93.9 |
| Clinical BioBERT | 93.1 | 93.4 | 94.3 | 93.0 |
| UmlsBERT | 93.6 | 94.4 | 94.9 | 94.3 |
| CODER | 98.0 | 96.9 | 97.7 | 97.8 |
| BGE | 97.2 | 96.1 | 97.1 | 97.1 |
| LINE | **98.1** | **97.2** | **98.1** | **98.1** |

tasks are considered: (1) a zero-shot clinical concept similarity evaluation task; (2) two standard i2b2 clinical NLP tasks; and (3) a renal cancer recurrence detection task.

### 3.3.1 Clinical concept similarity

We evaluated the quality of the LINE-projected embeddings based on their ability to detect known relationships among 222K pairs of 159K concepts. These pairs were curated from several major UMLS relation classes, including "may treat or may prevent", "classifies", "differential diagnosis", "method of", and "causative". Detailed counts of positive pairs for each class are provided in the last row of Table 2. To ensure a fair comparison with the other models, we removed the edges corresponding to these evaluation pairs from the relational adjacency matrices used in the LINE model. We then assessed the model's ability to capture different types of relationships by calculating cosine similarities between the embedding vectors of related pairs and comparing these to randomly selected pairs with similar semantic characteristics. For example, when evaluating the "may treat or may prevent" relationship, we used related disease-drug pairs as positive examples and compared them against negative pairs also composed of disease-drug pairs.

We calculated the Area Under the Curve (AUC) of the cosine similarities to assess the models' ability to distinguish known pairs from random ones. As shown in Table 2, GPT-4 and BGE outperform CODER and all other baseline clinical LLMs, serving as strong benchmarks. Notably, the LINE-projected embeddings achieve AUCs that not only exceed those of the original CODER model but also outperform the BGE benchmark in six of the seven relation classes and the GPT-4 benchmark in four of the seven. In the remaining classes, LINE's AUCs are comparable to their respective BGE and GPT-4 benchmarks, with a deviation less than 0.008.

Table 4: Performance metrics of sentence embeddings and their proxy embeddings generated from concept lists for various models, averaged over five-fold cross-validation. Improvements over the corresponding teacher models are indicated by "↑".

| Model | Concept | | | | Sentence | | | |
|---|---|---|---|---|---|---|---|---|
| | Precision | Recall | F1 | Accuracy | Precision | Recall | F1 | Accuracy |
| BioBERT | 0.701 | 0.686 | 0.693 | 0.701 | 0.681 | 0.669 | 0.675 | 0.680 |
| Clinical BioBERT | 0.722 | 0.723 | 0.722 | 0.726 | 0.726 | 0.726 | 0.711 | 0.729 |
| SapBERT | 0.726 | 0.711 | 0.718 | 0.722 | 0.741 | 0.723 | 0.732 | 0.722 |
| PubMedBERT | 0.715 | 0.697 | 0.706 | 0.708 | 0.725 | 0.711 | 0.718 | 0.708 |
| CODER | 0.711 | 0.691 | 0.701 | 0.708 | 0.707 | 0.706 | 0.706 | 0.698 |
| BGE | 0.728 | 0.695 | 0.711 | 0.711 | 0.778 | 0.768 | 0.773 | 0.764 |
| GPT-4 | 0.718 | 0.687 | 0.694 | 0.711 | 0.805 | 0.787 | 0.791 | 0.781 |
| LINE | | | | | | | | |
|   BGE+CODER | 0.741 ↑ | 0.728 ↑ | 0.734 ↑ | 0.722 ↑ | 0.778 | 0.765 | 0.771 | 0.771 ↑ |
|   GPT-4+CODER | 0.731 ↑ | 0.715 ↑ | 0.714 ↑ | 0.722 ↑ | 0.806 ↑ | 0.786 | 0.789 | 0.781 |

Table 5: Difference in performance metrics between sentence embeddings and their proxy embeddings generated from concept lists, calculated using results in Table 4. Reductions in difference, which indicate improved alignment, are marked by "↓".

| Model | Precision | Recall | F1 | Accuracy |
|---|---|---|---|---|
| BGE | 0.050 | 0.073 | 0.062 | 0.053 |
| GPT-4 | 0.087 | 0.100 | 0.097 | 0.070 |
| LINE | | | | |
|   BGE+CODER | 0.037 ↓ | 0.037 ↓ | 0.037 ↓ | 0.049 ↓ |
|   GPT-4+CODER | 0.075 ↓ | 0.071 ↓ | 0.075 ↓ | 0.059 ↓ |

### 3.3.2 Clinical NLP benchmarks

We evaluated our model on two standard biomedical named entity recognition (NER) benchmark tasks: the i2b2 2006 de-identification challenge [30] and the i2b2 2014 de-identification challenge [26]. We followed the train/validation/test splits specified in the original challenges, as detailed in Table 1 of [2]. The datasets from 2006 and 2014 contain 317 and 43 label classes, respectively. The NER tasks involve tokenizing sentences and then classifying each token within the sentence. For this task, we used token-level embeddings from each model and the LINE projection was applied to token-level BGE embeddings. GPT-4 was not included due to the limited access to its token-level embedding. Our fine-tuning process followed the setting in [15]. Specifically, a single linear layer was added on top of each model and trained for 20 epochs. We adopted Adam optimizer and the learning rates for CODER, BGE and LINE were set to $2 \times 10^{-5}, 2 \times 10^{-5}$ and $5 \times 5 \times 10^{-4}$, respectively. The mean of the F1 scores calculated over five random initializations are reported in Table 3, and it can be observed that LINE achieves the best overall performance.

### 3.3.3 Renal cancer recurrence detection

The dataset used for this downstream task comes from Mass General Brigham and was previously curated, fully de-identified, and annotated for another ongoing project. Here, we briefly describe the curation and annotation process. First, a phenotyping algorithm [37] was applied to identify a pool of renal cell carcinoma (RCC) patients. From this pool, 300 RCC patients were randomly selected, and a clinician reviewed all notes to extract one diagnostic-relevant sentence per patient. Each sentence was annotated into one of three classes: uninformative of RCC recurrence, informative of past or current recurrence, and informative of no recurrence. We then used NILE [36] to extract clinical concepts from each sentence, excluding those where NILE failed to extract any concept. This resulted in a final set of 288 labeled sentences, each paired with a list of corresponding clinical concepts. The dataset comprises 136 uninformative sentences, 64 informative of recurrence, and 88 informative of no recurrence.

This classification task is designed to test how effectively proxy sentences, generated from concept lists, can detect recurrence information. Specifically, we aim to understand how well the proxy sen-

tences perform in classifying uninformative, recurrence-informative, and non-recurrence-informative cases, simulating scenarios where only concept lists, not full sentences, are available. To this end, we additionally include classification results using raw sentences as a benchmark for expected optimal performance. We used LINE-projected CODER to embed the concept lists and LINE-projected GPT-4 or BGE to embed the raw sentences. A single linear layer was added on top of both models, and we fine-tuned it for 2000 epochs, with early stopping when the change in loss was less than $10^{-5}$. We used Adam optimizer with a learning rate of $10^{-3}$. The same setup was applied to all models for fair comparison. On average, the fine-tuning process took approximately one or two minutes per model. The average performance metrics from five-fold cross-validation are reported in Table 4. As shown, when raw sentences are available, GPT-4 and BGE achieve significantly better classification results than the other benchmarks. LINE-projected embeddings for GPT-4 and BGE manage to maintain a comparable level of classification performance. However, using only concept lists for GPT-4 and BGE results in a substantial drop in classification performance. Notably, adding the LINE projection reduces this performance gap, as shown in Table 5, supporting the effectiveness of the LINE module in better approximating sentence embeddings using concept lists.

## 4 Conclusion and Discussion

This paper introduces a teacher-teacher paradigm in which two pretrained LLMs align different forms of data to enable knowledge exchange and cross-form representation through a two-stage, few-epoch training of the LINE module, guided by a well-defined alignment objective. Our downstream analysis demonstrates that (1) the proposed LINE module effectively generates cross-form representations, and (2) alignment learning enhances performance for both models, even when one teacher model is more advanced than the other.

Although our primary focus is on a specific clinical use case, this teacher-teacher framework has broad potential applications. One promising application is enhancing the indexing and searchability of non-textual data. For example, medical images (e.g., CT or MRI scans) are often accompanied by clinical notes from which clinical concepts can be extracted. By segmenting images into regions and associating them with clinical concepts, we create a concept-based index of image content. The teacher-teacher framework aligns each image segment with its concept list, allowing the concept list to serve as a proxy representation for image content. This enables concept- or keyword-based searches for specific image segments via embeddings, making non-textual data more searchable. The framework also supports alignment between modalities without direct pairing. For example, CT and MRI scans are seldom captured together for the same patient, but clinical notes serve as a potential intermediary. In this scenario, clinical concepts extracted from notes are embedded by a pretrained teacher model. Separate teacher models then embed CT and MRI images. The alignment objective splits into two tasks: aligning clinical concepts with CT images and aligning clinical concepts with MRI images. This dual alignment establishes an indirect connection between CT and MRI images, creating a unified representation without direct pairing. These examples illustrate the framework's broad applicability in aligning disparate modalities, supporting cross-form searches and enhancing the utility of diverse data sources.

Our clinical use case further demonstrates the practical benefits of this framework in improving patient care and data security. By reducing the need for direct access to clinical notes, the framework minimizes privacy concerns and strengthens data protection. Additionally, it facilitates the integration of up-to-date LLMs, improving performance in tasks essential to patient care and potentially leading to better outcomes. The framework's use of external, factual knowledge also reduces misinformation risks, fostering trust in AI-assisted clinical tools.

Despite its strengths, the framework has two notable limitations. First, its reliance on paired data may limit its applicability in situations where aligned datasets are scarce. Although intermediaries can help bridge certain gaps, the framework's effectiveness remains constrained in cases where no suitable intermediaries are available. Second, the framework requires access to a comprehensive knowledge graph for effective training. In cases where such resources are unavailable or incomplete, model performance and generalizability may be impacted.

## Acknowledgment

We are deeply grateful to the area chair and the anonymous reviewers for their valuable comments, which greatly improved the quality of this paper.

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

# A  Appendix

## A.1  Handling negation in clinical concepts

Here, we provide additional details on how we handle concepts that are negatively mentioned. For example, consider the sentence, "*There is no focal consolidation, pleural effusion, or pneumothorax.*" The terms "focal consolidation", "pleural effusion", and "pneumothorax" are stated as negative, indicating their absence. This negation significantly alters the sentence's meaning compared to a positive assertion, such as "*There is focal consolidation, pleural effusion, or pneumothorax.*" Therefore, these negative concepts must be treated differently. To address this, we propose creating a dictionary of negative concepts derived from the original positive concepts, treating them as distinct entities. For example, the positive concept "pneumothorax" would have a corresponding negative "concept pneumothorax unobserved". Since the Unified Medical Language System (UMLS) only includes relations for positive concepts, we update the negative concepts by first using the multihead graph attention module, as described in Section 2.3, to update their corresponding positive concepts. The negative concept embeddings then pass through a projection layer. To ensure differentiation, we introduce a loss function to maintain a cosine similarity between the updated negative concept and its corresponding positive concept below a predefined threshold, $\delta$. For each positive-negative concept pair, denoted as $(\mathbf{c}_p, \mathbf{c}_n)$, the corresponding loss is calculated as follows:

$$-\log \frac{e^{\delta - \cos(\mathbf{c}_p, \mathbf{c}_n)}}{1 + e^{\delta - \cos(\mathbf{c}_p, \mathbf{c}_n)}},$$

with $0 < \delta \leq 0.5$. This loss function is implemented throughout our two-stage training process.

## A.2  Information recovered from residuals

In Table A1, we present some examples of crucial clinical concepts that remain undetected due to misspellings but can be subsequently recovered in the first stage of our training process.

Table A1: Examples of crucial clinical concepts that remain undetected due to misspellings but can be subsequently recovered through our two-stage training process.

| Original Sentence | Undetected Concept | Recovered Concept |
|---|---|---|
| Nasal fractureEpistaxisNSTEM | fracture Epistaxis NSTEM | Fracture |
| year old man with traumatic hemothorax s/p pigtail drainageand removal// please assess for interval change s/p pigtail removal****Pleaseperform at 3:30PM**** | drainage | External drainage |
| Hypodense appearance of the anterior rightlobe of the liver near the falciform ligament is likely perfusional. | right lobe | Right |
| Pulmonary markings are likely accentuated by lower lung volumes. | accentuated | Somewhat Worse |
| Globes andlenses are intact. | lenses | Cornea |

Table A2: Further results for Table 1.

| Models / Metric | Mean Rank ↓ | Mean Reverse Rank ↑ | Top10@Acc ↑ |
|---|---|---|---|
| CODER→BGE | 2.509 | 0.732 | 0.968 |
| BGE→CODER | 3.281 | 0.648 | 0.942 |
| LINE | | | |
|   BGE+CODER | 1.437 | 0.873 | 0.997 |

Table A3: Further results for Table 4.

| Model | Concept | | | | Sentence | | | |
|---|---|---|---|---|---|---|---|---|
| | Precision | Recall | F1 | Accuracy | Precision | Recall | F1 | Accuracy |
| CODER→BGE | 0.690 | 0.667 | 0.666 | 0.670 | 0.670 | 0.667 | 0.656 | 0.663 |
| BGE→CODER | 0.702 | 0.687 | 0.685 | 0.695 | 0.783 | 0.766 | 0.769 | 0.767 |
| LINE | | | | | | | | |
|   BGE+CODER | 0.741 | 0.728 | 0.734 | 0.722 | 0.778 | 0.765 | 0.771 | 0.771 |

## A.3 Comparison with related literature

In this subsection, we compare the proposed teacher-teacher framework with Retrieval-Augmented Generation (RAG), a method that combines large language models with knowledge databases to improve generation quality. A key distinguishing feature of our framework is that, rather than directly incorporating a knowledge database into Teacher 1 to enhance generation, we introduce a second generative model, Teacher 2, which is inherently endowed with relational knowledge. Teacher 2 then transfers this knowledge to Teacher 1 through alignment learning. This structural difference also supports an additional objective of our framework: enabling Teacher 2 to generate a purely concept-based embedding that can serve as a proxy for the clinical text, i.e. cross-form generation.

## A.4 Further experiment results

We include two additional benchmarks to our validation task on the alignment objective (Section 3.2) and the renal cancer recurrence detection task (Section 3.3.3), as both tasks utilize two teacher models. Specifically, we projected the CODER embeddings onto the BGE embedding space (denoted as "CODER→BGE") and also performed the reverse projection from BGE to CODER (denoted as "BGE→CODER") using a projection matrix. Compared to the two benchmarks, the proposed LINE model achieves better alignment and reduces the performance gap between using sentences and only concepts to detect recurrence information.

Additionally, we report the standard deviation of the F1 scores over five random initializations on two i2b2 datasets in Table A4

## A.5 Computational efficiency

To demonstrate the computational efficiency of the proposed framework, using our training data, we compare the increase in training time over several training epochs for the following three approaches: (1) directly fine-tuning BGE or CODER, (2) fine-tuning with low-rank adaptation (LoRA [9]), and (3) training the LINE module while keeping the pretrained LLMs frozen. The computation time was estimated using the `tqdm` function on a single NVIDIA RTX 8000 GPU with 48GB of VRAM. The rough estimates, shown in Figure A1, indicate that even with LoRA, training large models like BGE takes several days per epoch on a single GPU, with increasing computational overhead as training progresses. In contrast, our proposed LINE module requires only 2 hours per epoch on the same hardware.

Table A4: The mean of the F1 scores for different models over five random initializations on two i2b2 datasets. The results for BioBERT, Clinical BioBERT and UmlsBERT were directly copied from [15] for comparison. Here, the LINE projection is applied to the token-level BGE embeddings. The best result under each metric is highlighted in bold font.

| Dataset | i2b2 2006 | | i2b2 2014 | |
|---|---|---|---|---|
| | Test. F1 | Val. F1 | Test. F1 | Val. F1 |
| BioBERT | $93.3 \pm 1.300$ | $93.8 \pm 0.300$ | $94.6 \pm 0.200$ | $93.9 \pm 0.500$ |
| Clinical BioBERT | $93.1 \pm 1.300$ | $93.4 \pm 0.200$ | $94.3 \pm 0.200$ | $93.0 \pm 0.300$ |
| UmlsBERT | $93.6 \pm 0.500$ | $94.4 \pm 0.200$ | $94.9 \pm 0.100$ | $94.3 \pm 0.500$ |
| CODER | $98.0 \pm 0.020$ | $96.9 \pm 0.030$ | $97.7 \pm 0.010$ | $97.8 \pm 0.010$ |
| BGE | $97.2 \pm 0.010$ | $96.1 \pm 0.040$ | $97.1 \pm 0.004$ | $97.1 \pm 0.030$ |
| LINE | $\mathbf{98.1} \pm 0.004$ | $\mathbf{97.2} \pm 0.007$ | $\mathbf{98.1} \pm 0.003$ | $\mathbf{98.1} \pm 0.010$ |

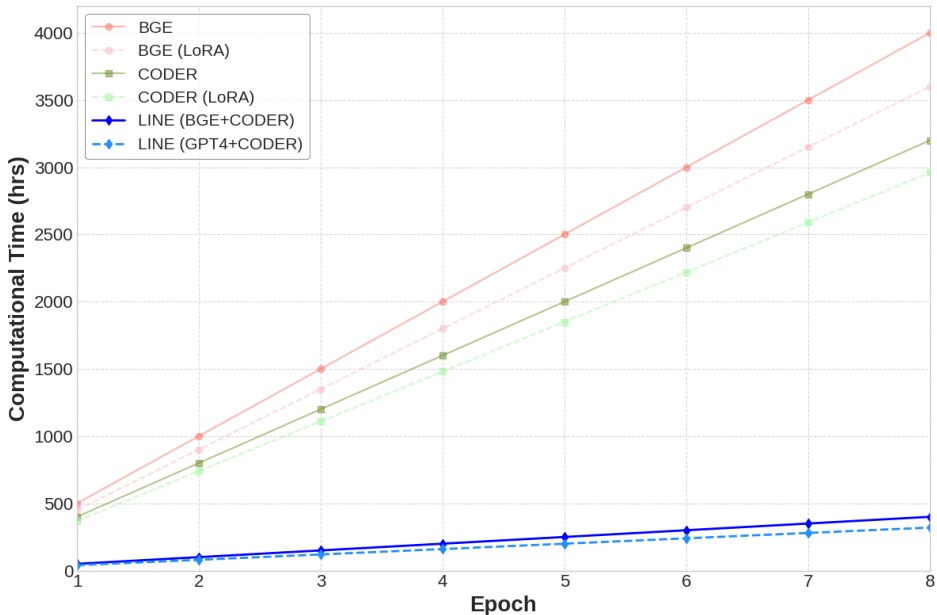

Figure A1: Computational Time Comparison between LINE and Baselines

