# OpenReview forum: "A teacher-teacher framework for clinical language representation learning"
_NeurIPS.cc/2024/Conference — NeurIPS 2024 poster_

### Official Review · Reviewer_j7j8 · 2024-06-15

**Soundness:** 3
**Presentation:** 3
**Contribution:** 3
**Rating:** 6
**Confidence:** 4

**Summary:**

The paper introduces a novel teacher-teacher framework named LIghtweight kNowledge alignmEnt (LINE), which facilitates knowledge exchange between two pre-existing large language models (LLMs) to enhance clinical language representation. By leveraging complementary knowledge from general-purpose and domain-specific models, LINE aims to harmonize their knowledge within a unified representation space. The framework is validated through downstream tasks showing that the LINE model outperforms individual pre-existing models in understanding and processing clinical language. This approach allows for more efficient sharing of clinical pretrianed models.

**Strengths:**

1. **Clarity and Structure**: The paper is well-written and structured, offering a clear motivation for the study. This makes it accessible and engaging for readers, facilitating a deeper understanding of the proposed framework.

2. **Novelty and Utility**: The proposed teacher-teacher framework, LIghtweight kNowledge alignmEnt (LINE), is innovative, providing a pragmatic approach to integrating the strengths of different pre-trained models. This methodology is particularly notable for its potential to enhance clinical language representations without the need for developing new models from scratch.

3. **Usability and Efficiency**: The framework is user-friendly and does not require retraining of the original models, which significantly reduces computational overhead and simplifies its adoption in real-world applications.

4. **Empirical Validation**: The experimental results demonstrate stable and significant improvements over existing methods, substantiating the efficacy and value of the proposed framework in practical settings.

**Weaknesses:**

**Data Requirements and Availability**: A notable limitation of the proposed LINE framework is its dependency on well-aligned and specific types of data sources, which may not be readily available or commonly found in practical settings. For example, integrating data from disparate modalities like CT and MRI requires the availability of cases that include both types of data, which may not always be feasible. This requirement could limit the framework's applicability across different clinical or real-world scenarios where such aligned data sets are scarce.

**Questions:**

1. See weakness, under such situation is it possible to apply your method?

**Limitations:**

No.
The authors discuss the potential improvement instead of the limitation of the current work. Bring more information and try other situations cannot be counted as an adequate discussion of limitations.

---

> ### Author Rebuttal · Authors · 2024-08-07
>
> > **Data Requirements and Availability**: A notable limitation of the proposed LINE framework is its dependency on well-aligned and specific types of data sources, which may not be readily available or commonly found in practical settings. For example, integrating data from disparate modalities like CT and MRI requires the availability of cases that include both types of data, which may not always be feasible. This requirement could limit the framework's applicability across different clinical or real-world scenarios where such aligned data sets are scarce. See weakness, under such situation is it possible to apply your method?
>
> This is an extremely insightful question, and we appreciate you bringing this up! Yes, our method can handle such situations by introducing an intermediary to bridge the gap between different data sources.
>
> For instance, if we want to infer $\mathbf{x}_1$ and $\mathbf{x}_3$ about each other but lack paired data $(\mathbf{x}_1, \mathbf{x}_3)$, the LINE framework can use an intermediary $\mathbf{x}_2$. This forms pairs $(\mathbf{x}_1, \mathbf{x}_2)$ and $(\mathbf{x}_2, \mathbf{x}_3)$, enabling the use of three teachers to jointly learn the alignment between all data pairs.
>
> In a concrete example involving CT and MRI scans, it is often rare for patients to have both scans simultaneously ($\mathbf{x}_1$ for CT and $\mathbf{x}_3$ for MRI). However, each type of scan typically comes with corresponding clinical notes ($\mathbf{x}_2$). These clinical notes can serve as intermediaries. By breaking down the clinical notes into concepts corresponding to specific scan regions, we can leverage Teacher 2, a pre-trained LLM, to incorporate factual relationships about these concepts. Teachers 1 and 3, pre-trained large vision transformers, can then embed the CT and MRI scans, respectively. The three teachers can train jointly, allowing the framework to effectively handle the alignment of disparate data modalities.
>
> >  The authors discuss the potential improvement instead of the limitation of the current work. Bring more information and try other situations cannot be counted as an adequate discussion of limitations.
>
> Thank you for your constructive comment. We recognize two main limitations of our framework:
>
> First, our approach requires data to come in pairs. While we can introduce intermediaries to create artificial pairs, this remains a limitation. The reliance on paired data may restrict the framework's applicability in scenarios where such pairs are not readily available.
>
> Second, effective training with a list of concepts necessitates a pre-existing factual knowledge graph. Access to such a knowledge graph is not always guaranteed, which can hinder the framework's performance and generalizability.
>
> We appreciate your feedback and will incorporate a more detailed discussion of these limitations in the paper.

---

> > ### Comment · Reviewer_j7j8 · 2024-08-11
> >
> > Thanks for your comment. I will keep my score.

---

### Official Review · Reviewer_Hfn8 · 2024-07-08

**Soundness:** 3
**Presentation:** 3
**Contribution:** 2
**Rating:** 4
**Confidence:** 4

**Summary:**

This paper presents an interesting topic on LLM but the importance of this problem is not convincing and the methods here is not novel.

**Strengths:**

The teacher-teacher concept is novel to some extent.

**Weaknesses:**

1. The problem's importance is not significant.
2. There lacks the inclusion of SOTA models like llama, gpt, etc.
3. The results improvement is limited as shown in Tab. 4,5.
4. The Fig1 lacks details of the proposed method.

**Questions:**

None.

**Limitations:**

See details in weakness.

---

> ### Author Rebuttal · Authors · 2024-08-07
>
> Thank you for your comment.We will address your comments point-by-point.
>
> >  The problem's importance is not significant.
>
> Thank you for raising this important question. While our motivating example originated from the medical domain, our LINE framework is broadly applicable to a wide range of scenarios. The core problem we address involves pairs of unlabeled data points $(\mathbf{x}_1, \mathbf{x}_2)$ from different domains, where $\mathbf{x}_2$ consists of items with known relationships derived from pre-existing factual knowledge graphs. Our objective is to learn an embedding $\mathbf{z}_2$ for $\mathbf{x}_2$ that closely matches the embedding $\mathbf{z}_1$ for $\mathbf{x}_1$. This general framework can be applied to various critical applications, including the alignment of text from structured EHR codes paired with clinical notes, images/audio paired with text. Below, we detail a few specific applications:
>
> 1. **Text Summarization**: In healthcare systems, clinical notes ($\mathbf{x}_1$) often need to be inferred from corresponding concepts ($\mathbf{x}_2$) due to regulatory constraints. Since raw clinical notes are frequently inaccessible, using a list of concepts to summarize and index notes enables de-identification and facilitates sharing among researchers. The concept list can be utilized instead of full clinical notes for downstream analysis.
> 2. **Text Summarization for Image and Audio**: In a cross-modality setting, the LINE framework can align audio/image data ($\mathbf{x}_1$) with corresponding textual information ($\mathbf{x}_2$). For instance, medical images are more challenging to anonymize than text due to embedded metadata and visual information that might indirectly reveal patient identity. In this context, LINE can employ a pre-trained vision transformer to embed images and a pre-trained language model to embed conceptual entities from clinical text. Consequently, LINE can provide embedding surrogates for images, such as CT scans, by leveraging only the concepts extracted from corresponding clinical notes. The text summaries can be used to both index the images and for direct downstream analyses. Additionally, our LINE algorithm has the potential to identify "residual" information in the images that is not captured by the paired data, which can then be projected into the text domain to provide meaningful insights.
>
> We will incorporate this into the motivation and discussion sections of the paper.
>
> > There lacks the inclusion of SOTA models like llama, gpt, etc.
>
> Thanks for your suggestion! During the rebuttal phase, we are able to use GPT4 as the strong teacher 1 to come up with LINE (GPT4+CODER). And then we further compared the new LINE results with the GPT4 baseline. The results are shown in the tables in the rebuttal pdf. We also provide a summary of the key results below for your easy reference. The following tables are additional results to be added to Tables 2, 3 and 5 of the paper. The better results are highlighted in bold.
>
> | Model / Metrics   | LINE (CODER+GPT4) | GPT4  |
> | ----------------- | ----------------- | ----- |
> | Mean Rank         | **1.477**         | 1.778 |
> | Mean Reverse Rank | **0.872**         | 0.820 |
> | Top10@Acc         | **0.995**         | 0.988 |
>
> | Models / Relation | Parent    | Sibling   | May Treat/Prevent | Classifies | DDX   | Method of  | Causative  |
> | ----------------- | --------- | --------- | ----------------- | ---------- | ----- | ---------- | ---------- |
> | GPT4              | 0.974     | **0.940** | 0.825             | **0.991**  | 0.939 | 0.934      | 0.935      |
> | LINE (GPT4+CODER) | **0.977** | 0.932     | **0.931**↑        | 0.988      | 0.938 | **0.965**↑ | **0.947**↑ |
>
> |                   | Concept   |           |           |           | Sentence  |           |           |           |
> | ----------------- | --------- | --------- | --------- | --------- | --------- | --------- | --------- | --------- |
> |                   | Precision | Recall    | F1        | Accuracy  | Precision | Recall    | F1        | Accuracy  |
> | GPT4              | 0.718     | 0.687     | 0.694     | 0.711     | 0.805     | **0.787** | **0.791** | **0.781** |
> | LINE (GPT4+CODER) | **0.731** | **0.715** | **0.714** | **0.722** | **0.806** | 0.786     | 0.789     | **0.781** |
>
> From the tables above, it can be observed that in most of the setting, LINE model performs better than directly using GPT4.
>
> > The results improvement is limited as shown in Tab. 4,5.
>
> As discussed in our response to your first question, our evaluation of the proposed framework is not solely focused on outperforming the benchmark model. It also emphasizes the potential of using a list of concepts to summarize notes, which enables de-identification and facilitates sharing among researchers. In Table 2, we evaluate whether the summarization from the list of concepts can be effectively aligned with the clinical text through a rank-based retrieval task. In Table 5, we assess whether the performance using only concepts can be comparable to the performance using raw text. The results from both the tables in the paper and the additional results presented above consistently demonstrate that the LINE model is more effective in summarizing text.
>
> > Figure 1 lacks details of the proposed method.
>
> Thank you for your comment. We have revised Figure 1 to include more details, which can be seen in Figure 1(b) of the rebuttal PDF. Specifically, Teacher 2 utilizes the trainable LINE module to first summarize the set $\mathbf{x}_2$, guided by pre-existing knowledge (i.e., the graph attention module and relational contrastive loss). Then, Teacher 2 generates an embedding $\mathbf{z}_2$ for $\mathbf{\tilde{x}}_2$ that closely resembles the embedding $\mathbf{z}_1$ for $\mathbf{x}_1$ through the alignment loss. Simultaneously, Teacher 1, a stronger LLM, uses a fully connected layer to learn the relational knowledge provided by Teacher 2 via the alignment loss.

---

> > ### Author Response · Authors · 2024-08-09
> >
> > ## Additional Experiment
> >
> > Thank you once again for your suggestion and thank you for your patience! We have incorporated an additional benchmark dataset focused on the mental health domain to further evaluate the proposed LINE framework against other methods. Specifically, we retrieved suicide-related publications from PubMed, using the search term "suici" to capture both "suicide" and "suicidal" references. This resulted in a collection of 8,000 publications with "suicid" in the title. We excluded a small number of large files (>215M) that required more than 16GB of RAM, accounting for approximately 1% of the total dataset.
> >
> > From these publications, we extracted titles and keywords, and further refined the dataset by removing any publications that did not contain keywords. We then conducted a rank-based retrieval task, using the keywords to retrieve the corresponding article. For each positive (title, keyword) pair, we generated 100 negative pairs by randomly substituting the title with one from a different article.
> >
> > Next, we computed the cosine similarity between the mean embedding of the keyword list and the title embedding for all pairs. These pairs were then ranked based on their cosine similarity scores, from highest to lowest. To assess performance, we calculated the mean rank, mean reverse rank, and Top-10 accuracy for all positive pairs. The results, presented in the table below, show that both LINE models show significant improvements over their respective teacher models.
> >
> > | Metric                | PubmedBERT | BioBERT | SapBERT | CODER | BGE  | CODER$\to$BGE | BGE$\to$CODER | GPT      | LINE(BGE+CODER) | LINE(GPT+CODER) |
> > | --------------------- | ---------- | ------- | ------- | ----- | ---- | ---------- | ---------- | -------- | --------------- | --------------- |
> > | **Mean Rank**         | 29.37      | 35.84   | 8.52    | 22.89 | 3.03 | 17.32      | 41.55      | 2.95     | 2.07            | **1.94**        |
> > | **Mean Reverse Rank** | 0.17       | 0.14    | 0.52    | 0.27  | 0.84 | 0.33       | 0.08       | **0.85** | 0.83            | **0.85**        |
> > | **Top10@Acc**         | 0.34       | 0.27    | 0.79    | 0.46  | 0.94 | 0.56       | 0.18       | 0.95     | 0.97            |        **0.98**         |

---

> ### Comment · Area_Chair_dR5b · 2024-08-11
>
> Dear Reviewer Hfn8,
> I am a NeurIPS 2024 Area Chair of the paper that you reviewed.
>
> This is a reminder that authors already left rebuttals for your review.
>
> We need your follow up answers on that. Please leave comment for any un-answered questions you had, or how you think about the author's rebuttal.
> The author-reviewer discussion is closed on Aug 13 11:59pm AoE.
>
> Best regards,
> AC

---

### Official Review · Reviewer_TNnd · 2024-07-11

**Soundness:** 3
**Presentation:** 3
**Contribution:** 3
**Rating:** 3
**Confidence:** 4

**Summary:**

The authors look to address the question representational alignment between language models trained on different textual domains to improve performance of potentially both models on their out-of-domain text. The authors propose to specifically investigate this in the context of EHR text, and choose as their models for this CODER and BGE. They propose a contrastive loss, and additionally propose to train an alignment module/project layer rather than end-to-end training of the teacher models.

**Strengths:**

The concept is solid and well implemented and motivated. I wonder if it would be possible to further generalize it beyond medical text - which it is restricted too due to the reliance on alignment with extracted medical concepts by NILE. The discussion mentions this possibility, but it would be exciting to see it in action.

The clinical NLP benchmarks are particularly appropriate for the task.

**Weaknesses:**

Some of the benchmark tasks are older, and the comparisons could be more robust. Some ablations are missing.

The project's scope is incredibly narrow: encoder models on extractive medical tasks. While the authors claim that the technique is broadly generalizable, it would be nice to see proof-of-concept.

The work seems to me to fit more into the realm of domain adaptation rather than learning by alignment. We aren't learning novel models here via alignment (like CLIP), but rather, pushing the learned representations of two different models into a common space. I'd strongly consider citing and discussing DA literature for this paper.

**Questions:**

Would it be possible to further test the LINE model on other, more varied, benchmarks to see how well those newly aligned representations perform?

It could also be exciting to explore this with generative models.

Were alternative frameworks considered for the concept alignment? Why not align directly in embedding space without the grounding concepts? This would be an interesting ablation to perform to assess the significance of the extracted concepts on the underlying learned representation. Conversely, could you just fine-tune the generalist model on the extracted concepts as a means of medically aligning it? How well does that perform?

Why not also compare the BGE-->CODER projection (inverse direction of the BGE-->CODER projection)?

If it isn't technically feasible to do in an end-to-end fashion, perhaps this could be approximated by tuning LoRA on the base models?

**Limitations:**

No discussion of limitations. Without additional experiments at the minimum a stated limitation should be the highly restricted domain of application (purely encoder models on medical topics).

---

> ### Author Rebuttal · Authors · 2024-08-07
>
> Thank you for your detailed comments. In the following, we will address your questions point-by-point.
>
> > The project's scope is incredibly narrow.
>
> Thank you for raising this important question. While our motivating example comes from the medical domain, our LINE framework is broadly applicable to various scenarios. The core problem we address involves pairs of unlabeled data points $(\mathbf{x}_1, \mathbf{x}_2)$ from different domains, where $\mathbf{x}_2$ consists of items with known relationships derived from factual knowledge graphs. Our goal is to learn an embedding $\mathbf{z}_2$ for $\mathbf{x}_2$ that closely matches the embedding $\mathbf{z}_1$ for $\mathbf{x}_1$. This framework can be applied to multiple critical applications, such as:
>
> 1. **Text Summarization**: In healthcare, clinical notes ($\mathbf{x}_1$) often need to be inferred from concepts ($\mathbf{x}_2$) due to regulatory constraints. Using a list of concepts to summarize notes enables de-identification and facilitates sharing among researchers. These concept lists can replace full clinical notes for downstream analysis.
> 2. **Text Summarization for Image and Audio**: The LINE framework can align audio or image data ($\mathbf{x}_1$) with textual information ($\mathbf{x}_2$). Medical images are harder to anonymize than text due to metadata and visual information. LINE can use a pre-trained vision transformer to embed images and a language model to embed conceptual entities from clinical text, providing embedding surrogates for images like CT scans. These text summaries can index the images and be used for downstream analyses. Additionally, LINE can identify "residual" information in the images not captured by paired data and project it into the text domain for insights.
>
> This discussion will be included into the paper. While including proof-of-concept on other settings would be beneficial, we are unable to do so due to time constraints.
>
> > I'd strongly consider citing and discussing DA literature for this paper.
>
> Thank you for your suggestion. We agree that our work is related to domain adaptation (DA) and will cite relevant literature, though our approach differs in key ways:
>
> - **Label Requirements**: Unlike DA, our method does not require labels during training. DA typically needs task-related labels for the source domain and may use "pseudo" labels for the target domain.
> - **Information Retention**: Our goal is to align latent embeddings to preserve both overlapping and complementary information, using a residual alignment training step to refine the alignment with additional concepts. DA focuses only on retaining overlapping information.
> - **Embedding Richness**: We aim to create semantic-rich embeddings by leveraging a graph-attention module, preserving relational information and mitigating rank degeneracy. DA often produces domain-agnostic embeddings, whose information richness depends on the specific task.
>
> We will further include this discussion in the paper.
>
> > Would it be possible to further test the LINE model on other, more varied, benchmarks?
>
> Thank you for your suggestion! In response, we are currently running an additional benchmark to evaluate the performance of the LINE models. We anticipate being able to present these results in the early stages of the discussion phase.
>
> >  It could also be exciting to explore this with generative models.
>
> The LINE framework generates in the embedding space, not the original data space. Given an unlabeled data pair $(\mathbf{x}_1, \mathbf{x}_2)$ from spaces $\mathcal{X}_1$ and $\mathcal{X}_2$, respectively, LINE aligns their latent embeddings $\mathbf{z}_1 \approx \mathbf{z}_2$. For a new observation $\mathbf{x}_2^{\text{new}}$, we can map it to the latent space to generate $\mathbf{z}_1^{\text{new}} \approx \mathbf{z}_2^{\text{new}} = \text{LINE}(f({\text{teacher 2}}(\mathbf{x}_2))$ for the missing $\mathbf{x}_1^{\text{new}}$. To extend this, we could add a decoder to map embeddings back to the clinical note space using a simple reconstructive loss. We will discuss this potential extension in future work.
>
> > Why not align directly in embedding space without the grounding concepts?
>
> Thank you for your thoughtful question. Grounding concepts in our framework both serve to integrate external factual knowledge to enhance model performance and safety, and also provide an independent data source for generating clinical note summaries, making them integral to our approach.
>
> > Why not also compare the BGE-->CODER projection?
>
> Thank you for the suggestion! Following your advice, we have included the results for the BGE-->CODER projection across our tasks in the rebuttal PDF. Overall, the projection between BGE and CODER yields worse results compared to the proposed LINE model.
>
> > Conversely, could you just fine-tune the generalist model on the extracted concepts as a means of medically aligning it?
>
> During the rebuttal phase, we attempted to fine-tune generalist models directly and with LoRA, but found it infeasible due to extremely long training times and high computational resource requirements.
>
> This highlights the computational efficiency of our proposed method. As shown in Figure 1(a) of the rebuttal PDF, even with LoRA, training the generalist BGE model takes at least 11 days per epoch, with rapidly increasing computational overhead. In contrast, our proposed model requires only about 2 hours per epoch using the same resources.
>
> > Without additional experiments at the minimum a stated limitation should be the highly restricted domain of application
>
> We acknowledge this concern and will include a discussion of the limitation that our current framework has only been tested on medical topics. We recognize the importance of evaluating our framework in other domains and will address this as a key area for future work.

---

> ### Comment · Area_Chair_dR5b · 2024-08-11
>
> Dear Reviewer TNnd,
> I am a NeurIPS 2024 Area Chair of the paper that you reviewed.
>
> This is a reminder that authors already left rebuttals for your review.
>
> We need your follow up answers on that. Please leave comment for any un-answered questions you had, or how you think about the author's rebuttal.
> The author-reviewer discussion is closed on Aug 13 11:59pm AoE.
>
> Best regards,
> AC

---

### Official Review · Reviewer_Yopj · 2024-07-13

**Soundness:** 2
**Presentation:** 3
**Contribution:** 3
**Rating:** 5
**Confidence:** 3

**Summary:**

This paper introduce a teacher-teacher framework for clinical language representation learning. The framework uses a lightweight knowledge alignment module to harmonize the knowledge of both models within a unified space, which including two steps: The first step involves initial training to define residuals and capture complementary information. The second step focuses on refining the alignment by recovering residual information. The framework was validated using the MIMIC-IV database, where the LINE model outperformed baseline models in aligning concept and text representations.

**Strengths:**

The main contribution of the work is proposed teacher-teacher framework, and training strategy.

- Originality: The teacher-teacher framework is very interesting as it enables mutual enhancement between two pre-existing LLMs, a unique departure from traditional approaches that typically involve training a new model or continual pre-training of existing models. This innovative method opens new avenues for leveraging existing resources to achieve superior performance.

- Quality: The paper demonstrates high quality through its validation using the MIMIC-IV database, a well-known and respected dataset in the clinical domain, adding significant credibility. Additionally, the LINE model's performance is compared against several strong baseline models, showing clear improvements across various downstream tasks, thus underscoring the robustness and reliability of the proposed framework.

- Clarity: The paper is well-written and clearly structured, making it accessible to both domain experts and those new to the field. The introduction provides a comprehensive background and motivation for the proposed framework, while the methodology section offers detailed descriptions of the teacher models and the LINE module.

- Significance: The practical applications and potential impact on the clinical domain shown the significance of this work. The teacher-teacher idea has substantial implications for more advancing NLP applications in other filed.

**Weaknesses:**

1. Figure 1 is somewhat confusing. From my understanding, Teacher 1 should be a strong LLM, while Teacher 2 should be an LLM with existing domain-specific knowledge. However, Figure 1 gives the impression that Teacher 2 serves merely as a database, making the framework resemble a RAG framework.

2. Although the paper compares the LINE model against several strong baseline models, it lacks a detailed comparison with the latest strong general LLMs, such as GPT-4, which should be considered a strong baseline. Consider adding a small comparative analysis or stating the advantages of the framework over simply using GPT-4.

3. The paper underscore the practical value of the framework, but it does not sufficiently address potential practical implementation challenges, such as computational requirements and scalability when applied in real-world clinical settings.

**Questions:**

1.From Figure 1, if Teacher 2 only serves to provide domain-specific knowledge, why not implement a RAG framework, which is training-free and potentially more reliable?

2. Have you addressed potential hallucination issues? Could one teacher potentially mislead the other during the knowledge exchange process?

3. What are the potential computational and scalability challenges of implementing the teacher-teacher framework in real-world clinical settings? How do you propose to mitigate these challenges?

4. How can regulatory mechanisms be incorporated into the framework for safety?

**Limitations:**

The paper has limited discussion on the broader implications of implementing the teacher-teacher framework in clinical settings. Consider to add the assessment of how the framework could impact patient care, data security, and trust in AI systems in healthcare.

---

> ### Author Rebuttal · Authors · 2024-08-07
>
> Thank you for your insightful comments! In the following, we will address them point-by-point.
>
> > Figure 1 is somewhat confusing. From my understanding, Teacher 1 should be a strong LLM, while Teacher 2 should be an LLM with existing domain-specific knowledge. However, Figure 1 gives the impression that Teacher 2 serves merely as a database.
>
> Thank you so much for pointing this out! We have modified Figure 1 into Figure 1(b) in the rebuttal PDF for better clarity. In summary, both Teacher 1 and Teacher 2 can generate embeddings when feeding in new data which are then aligned by LINE. When the data are paired, e.g. they come from the same hospital visit of a patient, their LINE-aligned embeddings will be close to each other.
>
> > From Figure 1, if Teacher 2 only serves to provide domain-specific knowledge, why not implement a RAG framework, which is training-free and potentially more reliable?
>
> The main role of Teacher 2 is to learn an embedding using a list of concepts to summarize the clinical notes, effectively matching the embedding generated directly from the raw notes. Since raw clinical notes are frequently inaccessible, summarizing and indexing notes using a list of concepts enables de-identification and facilitates sharing among researchers. Additionally, by allowing two teachers to align knowledge with each other via LINE, we hope to also improve the quality of the embeddings .
>
> While the RAG framework also leverages pre-existing knowledge networks to enhance learning, it is not a generative model and cannot be used to generate new embeddings. Therefore, it does not fulfill the same role as our proposed method, which not only aligns domain-specific knowledge but also generates new, robust embeddings.
>
> We will include RAG in our discussion of related work in the paper.
>
> > Although the paper compares the LINE model against several strong baseline models, it lacks a detailed comparison with the latest strong general LLMs, such as GPT4, which should be considered a strong baseline. Consider adding a small comparative analysis or stating the advantages of the framework over simply using GPT4.
>
> Thank you for your advice! During the rebuttal phase, we incorporated GPT4 as a strong Teacher 1 in our LINE framework, resulting in LINE (GPT4+CODER). We then compared the performance of this new model with the GPT4 baseline. The results, shown in the tables in the rebuttal PDF, indicate that LINE (GPT4+CODER) generally achieves better performance than GPT4 alone.
>
> > Have you addressed potential hallucination issues? Could one teacher potentially mislead the other during the knowledge exchange process?
>
> Thank you for the insightful question! To mitigate potential hallucination issues, we integrate factual knowledge from external reputable sources (e.g., UMLS) into our framework. Specifically, in our framework, external factual knowledge is incorporated into the training of Teacher 2 via the multihead graph attention module and the relational contrastive loss. This knowledge is then further propagated to Teacher 1 through the alignment loss.
>
> In this regard, Teacher 2 acts as a factually rigorous component that regularizes potentially misleading information from Teacher 1. We conducted an experiment to assess the quality of the LINE-aligned concept embeddings. The results, shown in Table 3 of the paper and the second table in rebuttal PDF, indicate that the generated concept embeddings faithfully preserve factual relationships and even slightly improve upon capturing these relationships compared to the baseline.
>
> > What are the potential computational and scalability challenges of implementing the teacher-teacher framework in real-world clinical settings? How do you propose to mitigate these challenges?
>
> Thank you for the question! One of the notable advantages of our proposed framework is its computational and memory efficiency, making it feasible to deploy with limited computational resources. To illustrate this, please refer to Figure1(a) in the rebuttal PDF, where we compare the training time of our model to both direct fine-tuning of LLMs and low-rank approximated fine-tuning of LLMs. The comparison shows that our model can be trained within two day on a RTX8000 48GB card.
>
> > How can regulatory mechanisms be incorporated into the framework for safety? Consider to add the assessment of how the framework could impact patient care, data security, and trust in AI systems in healthcare.
>
> Thank you for this important question! Below, we assess how the framework impacts patient care, data security, and trust in AI systems, and discuss the incorporation of regulatory mechanisms for safety. This discussion will be included in the paper.
>
> - **Data Security**: We utilize the publicly accessible MIMIC-IV dataset, which aligns with real-world clinical notes without privacy concerns. During inference, real medical notes aren't required; instead, a concept extractor tool like NILE can be used to extract key concepts for Teacher 2, preventing the use of sensitive patient information. Regulatory mechanisms can monitor and update the medical concepts in the knowledge graph, ensuring they reflect the most up-to-date factual knowledge. For instance, newly defined concepts like COVID-19 can be added, outdated concepts depreciated, and misleading concepts corrected.
> - **Patient Care**: The framework's computational efficiency allows for the effective use of state-of-the-art LLMs, improving downstream tasks such as disease diagnosis and lab results analysis, thereby enhancing patient care.
> - **Trust in AI Systems**: LINE can be retrained on local servers with various combinations of up-to-date LLMs, necessitating continual quality control. For example, our clinical concept similarity task (Section 3.3.1) can initially assess the fidelity of learned concept embeddings. During deployment, regular user feedback should be collected to further improve the system.

---

> ### Comment · Area_Chair_dR5b · 2024-08-11
>
> Dear Reviewer Yopj,
> I am a NeurIPS 2024 Area Chair of the paper that you reviewed.
>
> This is a reminder that authors left rebuttals for your review.
> We need your follow up answers on that. Please leave comment for any un-answered questions you had, or how you think about the author's rebuttal.
> The author-reviewer discussion is closed on Aug 13 11:59pm AoE.
>
> Best regards,
> AC

---

> > ### Comment · Reviewer_Yopj · 2024-08-12
> >
> > Thank you for the response! The new figure 1 makes sense to me. Thank you for new results. I decide to increase soundness from 2 to 3.

---

### Official Review · Reviewer_Yayo · 2024-07-14

**Soundness:** 3
**Presentation:** 3
**Contribution:** 3
**Rating:** 7
**Confidence:** 4

**Summary:**

The paper proposes a mutual learning framework, called LINE, between two pre-existing LLMs in the healthcare domains. By harmonizing the knowledge of two distinct LLMs into a unified representation space, the model achieves better performance on intrinsic and extrinsic downstream evaluations of clinical tasks.

**Strengths:**

Clear motivation. Overall well written.

The methodology was reasonably designed to map representations from two distinct LLMs into a unified representation space.

The method achieves better performance on downstream clinical tasks.

**Weaknesses:**

1. Only two LLMs (BGE and CODER) were aligned by LINE. It is unclear if LINE will work on combinations of other LLMs.

2. LINE make downstream predictions based on clinical concepts only, rather than the full context. The concepts themselves can be negated, historical and hypothetical in context, but the proposed method does not seem to consider this.

**Questions:**

1. Why was NILE selected? Have any other extractors been compared? Does the selection of extractors have a significant impact on results?

2. Line 222, which contrastive loss function was used eventually?

**Limitations:**

See weakness

---

> ### Author Rebuttal · Authors · 2024-08-07
>
> Thank you for your insightful comments! Below are our responses to your questions, addressed point-by-point:
>
> > It is unclear if LINE will work on combinations of other LLMs.
>
> Thank you for raising this important question. To address your concern, we have extended our experiments to include additional combinations of LLMs. Specifically, we tested the alignment of LINE with the combination of GPT4 and CODER. The results, presented in the tables in the rebuttal PDF, indicate that LINE (GPT4+CODER) generally achieves better performance than GPT4 alone.
>
> > The concepts themselves can be negated, historical and hypothetical in context, but the proposed method does not seem to consider this.
>
>  Thank you for your insightful comment! We agree that concepts can vary significantly depending on their context, and our approach does take some parts of this into account. Specifically, we have implemented mechanisms to handle negated concepts and will further include a discussion on the possibility of modeling for concepts in historical or hypothetical contexts.
>
> ##### Handling Negated Concepts:
>
> In our approach, we account for negated concepts by creating a separate dictionary of negative concepts derived from the original positive concepts. For instance, the positive concept "pneumothorax" is associated with a negated concept represented as "concept pneumothorax unobserved".
>
> To ensure that negated concepts are accurately represented, we use a multihead graph attention module (as detailed in Section 2.2) to update the embeddings of positive concepts, which are then adapted for their corresponding negative concepts through a projection layer. We introduced a loss function specifically designed to maintain a distinct representation for negated concepts. This loss function ensures that the cosine similarity between the updated negative $\mathbf{c}_p$ concept and its positive counterpart $\mathbf{c}_p$ remains below a predefined threshold, $\delta$, calculated as:
> $$
> -\log\frac{e^{\delta - \text{cos}(\mathbf{c}_p, \mathbf{c}_n)}}{1 + e^{\delta - \text{cos}(\mathbf{c}_p, \mathbf{c}_n)}},
> $$
> where $0 < \delta \leq 0.5$. This mechanism is detailed in the Appendix and has been implemented in our two-step training process.
>
> ##### Historical and hypothetical contexts:
>
> Additionally, we recognize the importance of considering whether concepts appear in historical or hypothetical contexts. Using NILE, we can effectively identify and extract contextually relevant keywords such as "previous", "since", or "in the near future". This capability enables us to incorporate context-specific modeling based on the temporal or hypothetical relevance of concepts. However, due to the complexity of real-world contexts, designing appropriate loss functions to account for these nuances requires further exploration and discussion.
>
> > Why was NILE selected? Have any other extractors been compared? Does the selection of extractors have a significant impact on results?
>
> We chose NILE primarily for its speed and convenience. Compared to cTAKES and MedTagger—both of which are popular extractors—NILE is about 2000 times faster than cTAKES and 400 times faster than MedTagger, while delivering comparable performance.
>
> Additionally, our framework includes a residual refinement step to recover important clinical concepts that might be missed due to issues like misspellings. This means that while the extractors need to perform reasonably well, they do not need to achieve extremely high accuracy. Consequently, the choice of extractor has a minimal impact on the overall results, as long as it effectively extracts crucial concepts.
>
> > Line 222, which contrastive loss function was used eventually?
>
> Thanks for allowing us to clarify! We use the triplet loss.

---

> ### Comment · Area_Chair_dR5b · 2024-08-11
>
> Dear Reviewer Yayo,
> I am a NeurIPS 2024 Area Chair of the paper that you reviewed.
>
> This is a reminder that authors left rebuttals for your review.
> We need your follow up answers on that. Please leave comment for any un-answered questions you had, or how you think about the author's rebuttal.
> The author-reviewer discussion is closed on Aug 13 11:59pm AoE.
>
> Best regards,
> AC

---

> > ### Comment · Reviewer_Yayo · 2024-08-11
> >
> > Thank you for the rebuttal! I will keep my recommendation unchanged.

---

### Author Rebuttal · Authors · 2024-08-07

Thank you all for your comments and questions! Based on your suggestions, we have made the following major changes during the rebuttal phase:

### Additional Experiment

1. **New Teacher Model**: We have adopted the OpenAI text embedding model "text-embedding-v3-small" as Teacher 1. Since it was released alongside GPT4 and demonstrates strong performance, we refer to it as "GPT4" for brevity. The subsequently trained model is referred to as "LINE (GPT4+CODER)". As suggested by the reviewers, we have included "CODER$\to$BGE" and "BGE$\to$CODER" as comparison benchmarks. "CODER$\to$BGE" projects the concept embeddings from CODER into the BGE embedding space using a projection matrix, while "BGE$\to$CODER" performs the inverse operation.

   The results are presented in the tables in the rebuttal PDF. Note that, since token-level embeddings from GPT4 are unavailable, we cannot perform i2b2 tasks using GPT4 or LINE (GPT4+CODER).

2. **Computational Efficiency**: To assess the computational efficiency of the proposed framework, we have benchmarked the estimated training time of our model against both direct fine-tuning of LLMs and low-rank approximated fine-tuning of LLMs [1] on a single RTX8000 48GB card. These results are shown in Figure 1(a) of the rebuttal PDF.

### Modified Figure 1 of the Paper

Following the reviewers' advice, we have modified Figure 1 in the paper for better clarity. Please see Figure 1(b) in the rebuttal PDF.

### Proposed Changes to the Paper

1. **Expanded Scope of Applicability**: We will add concrete examples to illustrate the broad applicability of the LINE framework.
2. **Additional Related Works**: We will include related literature on domain adaptation and retrieval-augmented generation to discuss how the proposed framework differs from these existing lines of research.
3. **Discussions on Safety**: We will add discussions on how the proposed framework addresses the issue of hallucination, manages concepts in different contexts, and assess its impact on patient care, data security, and trust in AI systems in healthcare.
4. **Discussion on Limitations**: We will add a discussion of the limitations of the proposed framework.

We appreciate your feedback and will make these changes to enhance the clarity and comprehensiveness of our paper.

[1] Hu, Edward J., et al. "Lora: Low-rank adaptation of large language models." *arXiv preprint arXiv:2106.09685* (2021).

---

### Decision · Program_Chairs · 2024-09-25

**Decision:**

Accept (poster)

**Comment:**

This paper introduces LINE framework, a novel teacher-teacher approach for clinical language representation learning. The paper presents an innovative method for harmonizing knowledge from two pre-existing large language models (LLMs) in the healthcare domain, resulting in improved performance on downstream clinical tasks.

The reviewers have highlighted several strengths of this work. The clear motivation and well-structured presentation make the paper accessible to both experts and newcomers in the field. The methodology is sound, demonstrating a unique approach to leveraging existing LLMs rather than training new models or continually pre-training existing ones. The use of the MIMIC-IV database for validation adds credibility to the results, and the performance improvements over strong baseline models underscore the robustness of the proposed framework.

During the rebuttal phase, the authors have addressed key concerns raised by reviewers. They expanded their experiments to include GPT4 as a teacher model, demonstrating that LINE (GPT4+CODER) generally outperforms GPT4 alone on various tasks. This addition significantly strengthens the paper by comparing against a state-of-the-art general LLM. The authors have also clarified their approach to handling negated concepts and provided insights into their choice of the NILE extractor.

The paper's weaknesses, such as the initial confusion in Figure 1 and the need for more detailed comparisons with recent LLMs, have been addressed in the rebuttal. The authors have committed to improving the figure and including additional experiments and discussions in the camera-ready version.

Given the innovative nature of the work, its potential impact on clinical NLP applications, and the authors' thorough responses to reviewer concerns, I recommend accepting this paper. The LINE framework presents a significant contribution to the field by introducing a novel approach to aligning and enhancing existing LLMs for clinical applications. The authors have demonstrated a clear understanding of the limitations and potential challenges of their approach and have shown commitment to addressing these in the final version.

In conclusion, this paper offers a technically sound and impactful contribution to the field of clinical NLP, with potential applications beyond healthcare. The authors' responsiveness to reviewer feedback and willingness to enhance the paper further strengthen its case for acceptance.